# MaskOpt: A Large-Scale Mask Optimization Dataset to Advance AI in Integrated Circuit Manufacturing

## Abstract

As integrated circuit (IC) dimensions shrink below the lithographic wavelength, optical lithography faces growing challenges from diffraction and process variability. Model-based optical proximity correction (OPC) and inverse lithography technique (ILT) remain indispensable but computationally expensive, requiring repeated simulations that limit scalability. Although deep learning has been applied to mask optimization, existing datasets often rely on synthetic layouts, disregard standard-cell hierarchy, and neglect the surrounding contexts around the mask optimization targets, thereby constraining their applicability to practical mask optimization. To advance deep learning for cell- and context-aware mask optimization, we present MaskOpt, a large-scale benchmark dataset constructed from real IC designs at the 45nm node. MaskOpt includes 104,714 metal-layer tiles and 121,952 via-layer tiles. Each tile is clipped at a standard-cell placement to preserve cell information, exploiting repeated logic gate occurrences. Different context window sizes are supported in MaskOpt to capture the influence of neighboring shapes from optical proximity effects. We evaluate state-of-the-art deep learning models for IC mask optimization to build up benchmarks, and the evaluation results expose distinct trade-offs across baseline models. Further context size analysis and input ablation studies confirm the importance of both surrounding geometries and cell-aware inputs in achieving accurate mask generation.

## 1 Introduction

In integrated circuit (IC) manufacturing, circuit patterns are transferred from a photomask to a silicon wafer using optical lithography (Chiu & Shaw, 1997). As the critical dimension of IC pattern continues to shrink, the nanoscale pattern has reached the limit of lithographic exposure wavelength, making precise wafer image printing increasingly challenging due to the optical diffraction effect and process variability. At this stage, resolution enhancement techniques (RETs) are employed to enhance the fidelity and printability of pattern transfer. Optical proximity correction (OPC) is a widely used RET. It refines the pattern shapes on a mask by compensating for the diffraction effect in the lithography process (Yu et al., 2023).

Model-based OPC (Awad et al., 2014; Kuang et al., 2015; Su et al., 2016; Matsunawa et al., 2015) and inverse lithography technique (ILT) (Poonawala & Milanfar, 2007; Gao et al., 2014; Jia & Lam, 2010) are prominent OPCs in the advanced node. Model-based OPC mathematically models the lithography process and move the edge segments iteratively to correct lithographic errors. ILT formulates the mask optimization as an inverse problem of the imaging system, optimizing an objective function to directly generate mask shapes. Compared to model-based OPC, ILT has the advantages of high imaging quality and a larger process window, owing to its pixel-based mask representation and global optimization approach (Yang et al., 2025). However, these methods are computationally intensive and require significant runtime. Since they take the printed contour shapes on the wafer as mask correction criterion, multiple rounds of lithography simulation are indispensable in the OPC flow, which substantially increases the computational complexity and runtime.

To improve the efficiency of mask optimization, cell-based hierarchical OPC (Wang et al., 2005; Sun et al., 2025; Yenikaya, 2017) has been widely adopted by leveraging the repetitive and modular

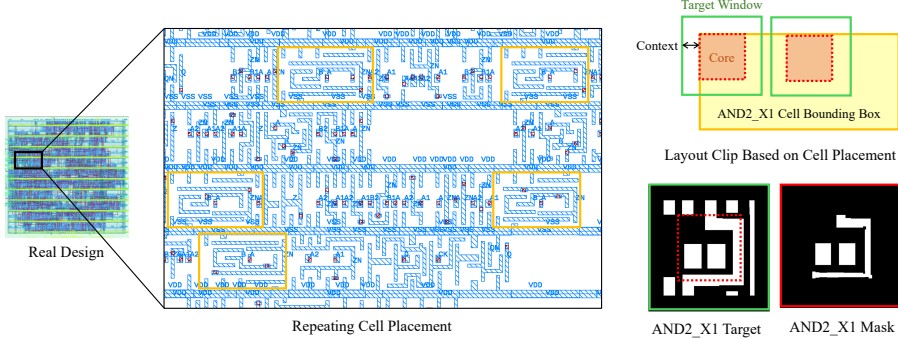

Figure 1: Visualization of real IC layout clipping for MaskOpt.

nature of IC designs. Instead of applying OPC across the full chip layout, hierarchical OPC optimizes individual cells (e.g., standard cells), which are then reused throughout the design, thereby achieving significant savings in runtime and computational resources (Pawlowski et al., 2007). By operating at the standard-cell level, mask optimization benefits from the repeated instances of common logic gates and functional blocks. However, the optimized mask of a cell is not universally applicable across the chip or even across chips fabricated at the same node, as variations emerge from optical proximity effects induced by neighboring geometries as well as chip-specific OPC calibrations. Although recent efforts have begun integrating artificial intelligence into OPC engines for improved scalability and accuracy, most deep learning (DL) approaches (Zheng et al., 2023a; Yang et al., 2018; Chen et al., 2020; Jiang et al., 2021) remain constrained by dataset design choices.

Existing mask optimization datasets such as LithoBench (Zheng et al., 2023a) primarily generate masks from isolated layout tiles, neglecting the hierarchical structure of real IC layouts. For metal layers, layout tiles are often synthesized based on simple design rules, which limits the ability of trained models to generalize to practical designs. For via layers, tiles are typically obtained by clustering via patterns by density or slicing real layouts with fixed strides, overlooking the repetitive via patterns across circuits. Moreover, optical lithography is inherently sensitive to optical proximity effects (Flagello et al., 2008), where printed images are influenced by the surrounding shapes around optimization target. As illustrated in Figure 1, the same standard cell can appear in diverse local contexts, and ignoring these environmental contexts prevents DL-based OPC models from capturing the variations of mask patterns in different placements.

To address these limitations, we introduce MaskOpt, a new benchmark for training and evaluating deep learning models for IC mask optimization. The dataset consists of layout tiles of standard cells extracted from real IC designs, paired with masks generated using advanced model-based OPC and ILT techniques (Zheng et al., 2023b) in academia. To preserve hierarchical cell information, as illustrated in Figure 1, each tile is clipped by applying a core region within the bounding box of a standard-cell placement, while the clipping window is extended with variable context sizes to capture the influence of neighboring features on mask patterns. We further evaluate recent state-of-the-art models on MaskOpt to establish baseline performance and provide a foundation for future research. Input ablation experiments show that removing cell information from model input degrades performance, especially on via layers and ILT tasks, underscoring the necessity of cell-aware inputs for accurate mask optimization. Context analysis reveals that incorporating surrounding geometries improves prediction accuracy, with metal layers performing best with small context and via layers benefiting from larger context due to their sparsity.

**Contributions.** (1) We introduce MaskOpt, a new large-scale IC mask optimization dataset designed to promote ML innovation for practical large-scale IC mask generation. The dataset is built from real IC designs and explicitly accounts for the varying placement contexts of standard cells. Each sample consists of a layout target and associated cell tag as input, with the corresponding model-based OPC and ILT masks as output for both metal and via layers. (2) We provide benchmarks by evaluating state-of-the-art deep learning model baselines for both model-based OPC and ILT mask generation tasks, establishing comprehensive model evaluations. (3) We empirically demonstrate the value of cell and context information for mask optimization. Input ablation experiments show mask generality quality improvements with cell-aware inputs across most tasks, while context analysis shows that leveraging surrounding patterns further enhances prediction accuracy.

## 2 RELATED WORK

### 2.1 OPTICAL LITHOGRAPHY

Optical lithography is a key technology in semiconductor manufacturing, enabling the transfer of intricate circuit patterns onto wafers with nanometer-scale precision (Okazaki, 1991; Mack, 2008). A lithographic system consists of three main components: (1) a light source, typically deep ultraviolet (DUV) or extreme ultraviolet (EUV) radiation, that provides illumination; (2) the mask, which encodes the designed circuit features; and (3) the photoresist, a light-sensitive layer on the wafer that records the projected pattern. After exposure, post-exposure bake, development, and etching processes transfer the recorded image into the underlying substrate, defining device features.

During the lithography process, mask $M(x, y)$ is illuminated and projected through the optical system to form the aerial image $I$, which represents the light intensity distribution on the resist surface. Hopkins diffraction (Hopkins, 1951; Cobb, 1998) has been widely applied to mathematically model the projection:

$$I(x, y) = \sum_{k=1}^{N} w_k \left| M(x, y) \otimes h_k(x, y) \right|^2, \tag{1}$$

where $h_k$ and $w_k$ are the $k^{th}$ kernel and its weight. After the optical projection, a photoresist model transfers the aerial image $I$ to the printed image $Z$, which can be modeled as:

$$Z(x, y) = \sigma_Z \big( I(x, y) \big) = \frac{1}{1 + e^{-\alpha(I(x,y) - I_{th})}}, \tag{2}$$

where $I_{th}$ is the intensity threshold, $\alpha$ is a constant number that controls the steepness of the sigmoid function.

### 2.2 OPTICAL PROXIMITY EFFECT

Optical proximity effects are distortions that occur when transferring a pattern from a photomask to a semiconductor wafer during optical lithography, preventing the final pattern from precisely matching the original design. As feature size shrinks to the scale of the exposure wavelength, the lithography system becomes diffraction-limited, meaning that the aerial image projected onto the wafer is fundamentally blurred.

The core of the proximity effect is that the aerial image of any given feature is dependent on its surroundings due to the superposition of diffraction patterns from all nearby features. In optical lithography, the aerial image of a target feature is determined by convolution of the mask with the imaging system's point-spread function, which has a finite extent determined by the numerical aperture and illumination wavelength (Mack, 2008). In OPC or ILT flows, a finite context window is clipped around target features to capture surroundings. At the 45nm node, proximity effects extend beyond immediate neighbors, often requiring $\sim 1\mu\mathrm{m}$ radius of influence in OPC kernels (Gupta et al., 2004).

### 2.3 MODEL-BASED OPC AND ILT

Target layout $Z_t$ denotes the intended circuit pattern to be transferred onto the wafer. To achieve a high-fidelity printing, the distortions between $Z$ and $Z_t$ caused by optical proximity effects must be counteracted. To address this, OPCs are employed to optimize the mask $M$ to improve printing fidelity. The widely used OPCs are model-based OPC and ILT.

**Model-based OPC.** Model-based OPC is performed by iteratively correcting specific features through online optical lithography simulation of the mask, described in Section 2.1. The flow of MB-OPC involves segmenting each feature into smaller fragments, simulating the aerial image and resist contour, comparing the simulated contour with the intended target, and applying localized corrections to each fragment. This iterative process continues until the corrected mask pattern produces a simulated wafer contour that closely matches the design specification.

**ILT.** ILT determines mask $M$ that produces the desired on-wafer results $Z$ by formulating the resolution enhancement process as an inverse problem of optical lithography, which can be modeled

as an optimization problem where the objective is to minimize a cost function that combines the $L_2$ norm between the printed and target patterns with additional mask evaluation metrics (detailed in Sec 4.1), such as the process variation band $L_{pvb}$ and edge placement error $L_{epe}$:

$$min_M(x,y) \|Z - Z_t\|_2^2 + L_{pvb} + L_{epe} \tag{3}$$

### 2.4 Deep Learning for Mask Optimization.

For IC mask optimization, deep learning methods provide significant speedups while maintaining mask printability by reducing reliance on slow, iterative simulations. GAN-OPC (Yang et al., 2018) uses generative adversarial networks (GAN) with ILT-guided pre-training to accelerate convergence, while DAMO (Chen et al., 2020) combines a high-resolution conditional GAN and a feed-forward network with back-propagated correction gradients to directly generate optimized masks. RL-OPC (Liang et al., 2024) frames mask optimization as a reinforcement learning problem, where an agent adjusts mask edges with lithography-driven rewards. Neural-ILT (Jiang et al., 2020) reformulates ILT into a neural network that jointly optimizes printability and shot count, simplifying mask patterns and reducing cost. CNFO (Yang et al., 2022) incorporates lithography physics into a Fourier neural operator for more accurate and data-efficient learning. EMOGen (Zheng et al., 2024) enables the co-evolution of pattern generation and ILT models, enhancing mask optimization via layout pattern generation. Since data scarcity remains a central challenge (Yang et al., 2022), recent work increasingly integrates physics into model design. For example, BSCNN-ILT (Chen et al., 2025) introduces a block-stacking framework with vector-based lithography physics modeling, enabling efficient mask optimization without large labeled datasets.

### 2.5 Mask Optimization Datasets

LithoBench (Zheng et al., 2023a) is the first benchmark dataset tailored for deep learning–based lithography simulation and mask optimization. It contains 16,472 synthesized tiles for metal-layer mask optimization, generated using the method from (Yang et al., 2019) under ICCAD-13 design rules (Banerjee et al., 2013) at the 32nm node, and 116,415 clipped tiles from OpenRoad-generated layouts at the 45nm node for via-layer mask optimization. ILT mask ground truths are produced using an advanced method derived from (Sun et al., 2023; Chen et al., 2023), which improves traditional pixel-based ILT through average pooling and multi-resolution schemes. While large in scale, LithoBench relies on synthesized layouts for metal-layer masks, limiting applicability to real designs. It also lacks key features such as cell information for cell-hierarchical OPC and contextual surroundings for target printing, both critical for accurate mask generation and model generalizability. Other related benchmarks, such as ICCAD-13 (Banerjee et al., 2013), which provides only 10 metal-layer tiles from 32nm industrial layouts, and GAN-OPC (Yang et al., 2018), which offers about 4k synthetic tiles, are too small to support deep learning training. To address these issues, we propose MaskOpt, a dataset for mask optimization with explicit support for cell-hierarchical OPC. Unlike prior datasets, MaskOpt is built from real IC designs across both metal and via layers, with ground-truth masks generated using widely adopted academic methods to ensure high-quality data.

## 3 The MaskOpt Dataset

### 3.1 Overview of MaskOpt

Our MaskOpt dataset has 104,714 metal-layer tiles and 121,952 via-layer tiles from five real IC designs at 45nm technology node. Unlike the prior dataset (Zheng et al., 2023a; Yang et al., 2018), which uses synthetic layout and clips layouts at the full-chip level, we emphasize cell-aware and context-aware mask optimization by clipping layout tiles with the bounding box of individual cell placement and extending the target with different context sizes. The statistical analysis of MaskOpt is shown in Fig. 2. MaskOpt includes layout tiles of logic gates belonging to standard cell families such as AND, AOI, BUF, NAND, NOR, OR, XNOR, XOR, and OAI. For each standard cell family, we clip layouts from its cell instances, covering different input configurations and drive strengths. For example, OAI22_X1 is an OR-AND-Invert (OAI) gate with two 2-input OR groups and the smallest drive strength. Since standard-cell layouts contain sparse via arrays, we collect via-layer data from all five designs to obtain a comparable number of examples to the metal layer. As shown

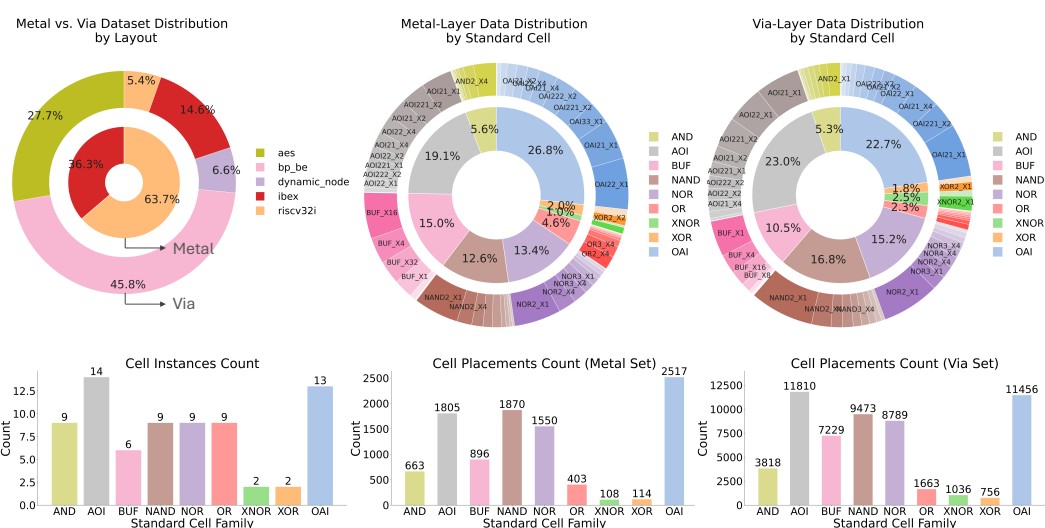

Figure 2: MaskOpt dataset statistics.

in Figure 3, each MaskOpt data example includes a target image of a layout tile with varying context sizes, its associated standard-cell tag, and two masks generated by model-based OPC and ILT methods. For mask optimization tasks, the inputs are the target image and its cell tag, and the output is the optimized mask.

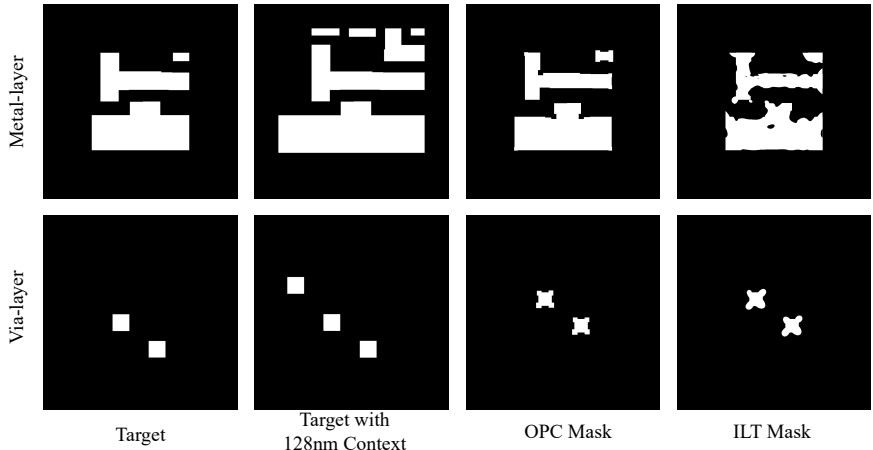

Figure 3: AOI221_X2 layout tile with mask samples.

## 3.2 DATA CURATION

**Circuit Designs.** We curate a collection of five real IC designs implemented with the Open-ROAD (Ajayi & Blaauw, 2019) flow using the Nangate 45nm open cell library. The design set spans diverse functional blocks, including encryption circuits, arithmetic units, and processor cores, offering greater diversity and practical relevance than the synthetic layouts provided in LithoBench.

**Layout Clip.** To enable cell-based hierarchical OPC, we clip layout tiles based on standard cell placements. We employ Algorithm 1 to generate layout tiles. Leveraging the repetitive nature of cell patterns, we identify the placement instances of each type of standard cell and sweep a core region of 512nm × 512nm within the instance's bounding box. Subsequently, we apply context margins of 0nm, 16nm, 32nm, 64nm, and 128nm around the core to crop the layout. We crop masks at the core coordinates. All layout tiles are converted to 1024 × 1024 target images with a spatial resolution of 1 pixel/nm$^2$.

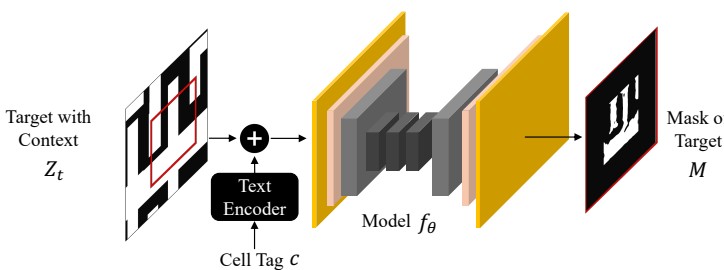

Figure 4: Mask prediction with cell and context awareness.

---

**Algorithm 1** Layout & Mask Clipping at Standard-Cell Placements

---

**Require:** Layout $L$; layers $\mathcal{L} = \{\text{metal}, \text{via}\}$; context size $S_c$; core size $S_{\text{core}} = 512\,\text{nm}$; standard-cell set $\mathcal{S}$; OPC masks $\{M_\ell^{\text{OPC}}\}_{\ell \in \mathcal{L}}$, ILT masks $\{M_\ell^{\text{ILT}}\}_{\ell \in \mathcal{L}}$

**Ensure:** Per-core outputs indexed by $t$: clipped windows $Z_t^\ell$, and core-cropped masks $C_{t,\ell}^{\text{OPC}}, C_{t,\ell}^{\text{ILT}}$

1:  Load layout $L$ and masks $\{M_\ell^{\text{OPC}}, M_\ell^{\text{ILT}}\}_{\ell \in \mathcal{L}}$
2:  **for all** cells $c$ in $L$ **do**
3:      **if** $c \in \mathcal{S}$ **then**
4:          $\mathcal{I} \leftarrow \text{INSTANCES}(L, c)$                   ▷ all placed instances of $c$
5:          **for all** instances $i \in \mathcal{I}$ **do**
6:              $B \leftarrow \text{BBOX}(i)$                  ▷ instance bbox in layout coords
7:              $\mathcal{K} \leftarrow \text{SWEEPCORE}(B, S_{\text{core}})$         ▷ pack core boxes inside $B$
8:              **for all** core boxes $K \in \mathcal{K}$ **do**
9:                  $W \leftarrow \text{EXPAND}(K, S_c)$     ▷ context expansion of the core on all sides
10:                 **for all** $\ell \in \mathcal{L}$ **do**        ▷ same cropping logic for metal & via
11:                     $\text{Shapes}_\ell \leftarrow \text{CLIP}(L, W, \ell)$    ▷ intersect layer $\ell$ with window $W$
12:                     **if** $\text{Shapes}_\ell \neq \emptyset$ **then**
13:                         Create new layout window $Z_t^\ell$
14:                         Insert $\text{Shapes}_\ell$ into $Z_t^\ell$
15:                         $C_{t,\ell}^{\text{OPC}} \leftarrow \text{CROPBYBOX}(M_\ell^{\text{OPC}}, K)$ ▷ mask crops use core coordinates
16:                         $C_{t,\ell}^{\text{ILT}} \leftarrow \text{CROPBYBOX}(M_\ell^{\text{ILT}}, K)$
17:                         $\text{SAVE}(Z_t^\ell, C_{t,\ell}^{\text{OPC}}, C_{t,\ell}^{\text{ILT}}, \text{meta}(i, c, K, W, \ell))$

---

**Mask Truth.** We employ model-based OPC and ILT methods in OpenILT platform (Zheng et al., 2023b) to generate mask ground truths. In OpenILT, model-based OPC segments polygon patterns into movable line segments, iteratively adjusts these edges according to simulated edge placement errors, and rasterizes the updated polygons into the final mask image. ILT optimizes the mask directly by minimizing a combination of $L_2$ loss and $L_{pvb}$ loss. To ensure realistic masks that capture the influence of sufficient surrounding context, we use an enlarged target window of $2048\text{nm} \times 2048\text{nm}$ for OpenILT simulation then clip the mask at the core region for the golden truth.

### 3.3 TASK SPECIFICATION

We formulate the task as predicting the model-based OPC and ILT masks of the core region in a target layout tile. Formally, as illustrated in Figure 4, given the layout target image with context $Z_t$ around the core and its associated cell tag $c$, the objective is to learn a model $f_\theta$ that outputs the mask $M$:

$$f_\theta(Z_t, c) \mapsto M, \tag{4}$$

where the parameters $\theta$ are optimized to minimize the loss $\mathcal{L}(f_\theta(Z_t, c), M^*)$ between the predicted mask and the ground-truth mask $M^*$ and additional metrics (detailed in Sec 4.1) such as the $L_2$ norm of $Z_t$ and printed image $Z$, process variation band $L_{\text{pvb}}$, and edge placement error $L_{\text{epe}}$.

### 4 EXPERIMENTS

### 4.1 EXPERIMENT SETUP

**Benchmark Models.** We benchmark four state-of-the-art opensource mask optimization models: GAN-OPC (Yang et al., 2018), Neural-ILT (Jiang et al., 2020), DAMO (Chen et al., 2020), and CFNO (Yang et al., 2022). All models are evaluated on ILT mask prediction. For OPC mask

prediction, we focus on GAN-OPC and DAMO, as these frameworks are inherently adaptable to both model-based OPC and ILT.

To enable cell- and context-aware mask optimization, we modify the generator structure of baseline models to incorporate the cell tag as an additional input. Since the cell tag is enumerable, we represent it as a one-hot encoded map instead of using a learned embedding. This one-hot representation is expanded to $1024 \times 1024$ and concatenated with the layout image along the channel dimension.

**Evaluation Metrics.** In mask optimization, the objective is not only to train a model that predicts accurate masks but also to ensure that the printed image of the generated mask after lithography simulation matches the target shapes. Therefore, we adopt the following commonly used metrics in mask optimization for the evaluation:

(1) *Square $L_2$ error*: Given the target layout image $Z_t$ and the printed wafer image $Z$ of a mask $M$, the squared L2 error is given by $\|Z - m_{\text{core}} \cdot Z_t\|_2^2$. Since the objective is to predict masks only within the core region, a binary zero-out mask $m_{\text{core}}$ is applied to the target layout $Z_t$, filtering out all non-core regions from the comparison.

(2) *Edge placement error (EPE)*: EPE refers to the vertical or horizontal misalignment, i.e., Manhattan distance from the lithography contour of $Z$ to the desired contour of the target pattern $m_{\text{core}} \cdot Z_t$. OpenILT toolkit samples points along target edges, and any point exceeding the predefined EPE constraint is counted as a violation. The total number of violations defines the EPE score.

(3) *Process Variation Band (PVB)*: PVB is defined as the area between the outermost and innermost printed edges across all process conditions, reflecting the robustness of a mask to process variations. In our experiments, PVB is measured under ±2% dose error and calculated as $|Z_{max} - Z_{min}|_2^2$, where $Z_{max}$ and $Z_{min}$ are the printed images under maximum and minimum process conditions.

(4) *Mask Fracturing Shot Count (#Shot)*: Given a mask $M$, the mask fracturing shot count denotes the number of rectangular shots for accurately replicating the mask shapes.

**Implementation Details.** We use KLayout (Köfferlein, 2020) to clip the designs and save the clipped tiles as GDS files, which are then converted into GLP format for OpenILT simulation. To obtain the printed wafer image $Z$ of predicted mask $M$, we employ the optical lithography simulation framework in OpenILT, which is from ICCAD-13 benchmark and applicable to 45nm technology nodes. All baseline models are implemented with PyTorch and trained on $2 \times$ NVIDIA A100 GPUs.

### 4.2 RESULTS

**Context Analysis.** In this section, we analyze the impact of context size on mask prediction accuracy. Baseline models are trained on different subsets of MaskOpt with varying input target context sizes. We report the MSE loss between the predicted masks and the ground truth masks.

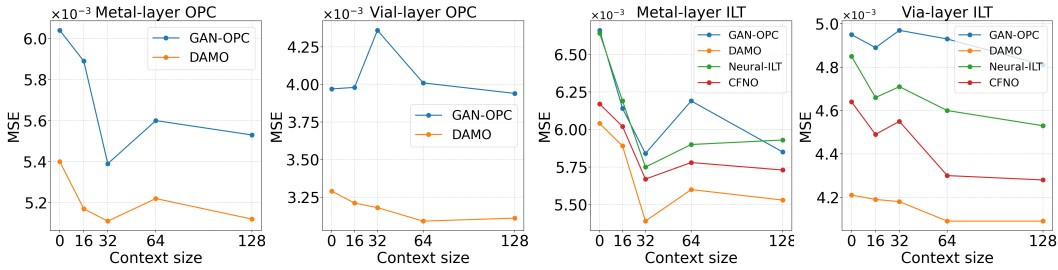

Figure 5: Context size analysis for mask prediction.

The results are presented in Figure 5. For both metal and via layers, incorporating surrounding shapes into the target input leads to improved mask generation, as demonstrated by consistent accuracy gains over the 0nm context setting across all baseline models. For metal-layer mask prediction, all models achieve their best accuracy with a relatively small context size of 32nm. In contrast, for

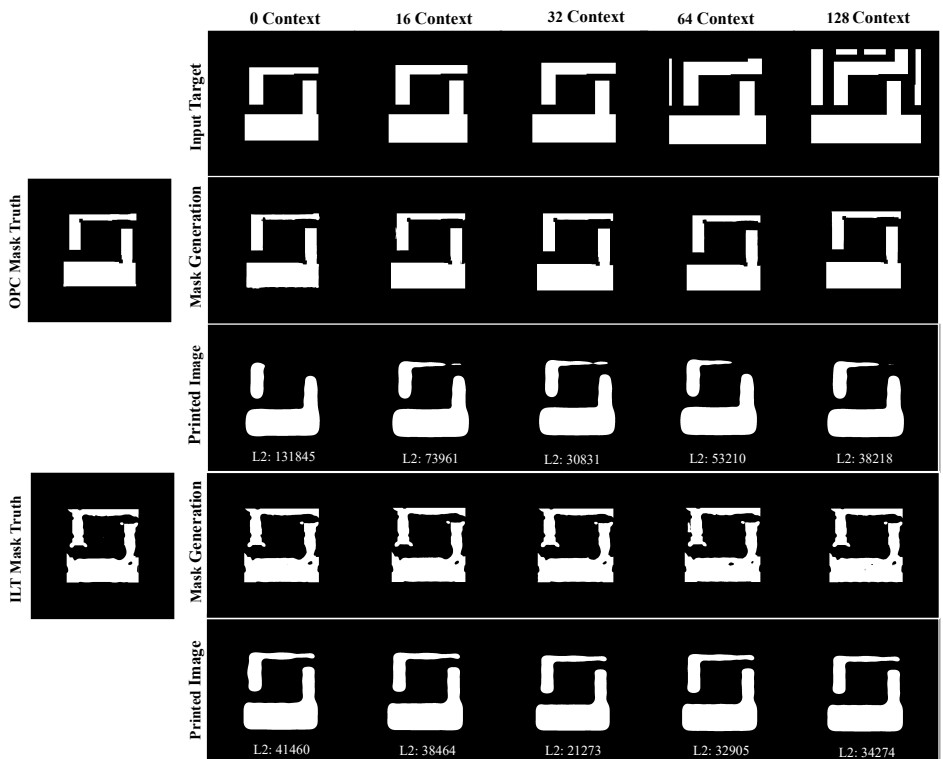

Figure 6: Mask prediction samples with different context sizes.

via-layer mask prediction, the highest accuracy is consistently obtained with the large context size of 128nm. We analyze that the difference arises from the sparsity of via patterns in standard cells, where larger context windows help capture neighboring shapes. Conversely, larger context windows for metal layers may introduce more complex geometries, potentially reducing prediction accuracy.

Figure 6 showcases predicted mask examples of AND2_X1 gate by OPC-GAN, along with corresponding printed images under different context sizes. We show the $L_2$ error to highlight the difference between the printed images and the target core. For both model-based OPC and ILT mask predictions, the lowest error is observed with a 32nm context, where the $L_2$ error reaches 30831 for the OPC task and 21273 for the ILT task.

Table 1: Quality Evaluation of Mask Generation. We use **bold** to indicate the best.

| Task | Model | Metal-layer | | | | Via-layer | | | |
|---|---|---|---|---|---|---|---|---|---|
| | | $L_2$ | EPE | PVB | Shot | $L_2$ | EPE | PVB | Shot |
| Model-based OPC | OPC-GAN | 58767 | 46.0 | 7051 | **153** | 18134 | 19.7 | 583 | **72** |
| | DAMO | **56076** | **44.4** | **6238** | 297 | **15922** | **19.6** | **551** | 96 |
| ILT | OPC-GAN | 60162 | 43.8 | 7699 | **634** | 17361 | **18.6** | **453** | 228 |
| | Neural-ILT | 59080 | 43.3 | **7614** | 665 | 17953 | 19.0 | 803 | 271 |
| | CFNO | 59781 | 44.9 | 7823 | 704 | 18980 | 19.2 | 627 | **221** |
| | DAMO | **56900** | **40.9** | 7773 | 740 | **18123** | 18.7 | 826 | 248 |

**Overall Results.** In this section, we evaluate the quality of masks generated by the baseline models. Training is performed independently on the metal-layer and via-layer subsets of MaskOpt, using a context size of 32nm for metal and 128nm for via. We report the mean $L_2$, EPE, PVB, and Shot metrics on the test set in Table 1. Figure 7 shows predicted AOI221_X2 mask samples of baseline models.

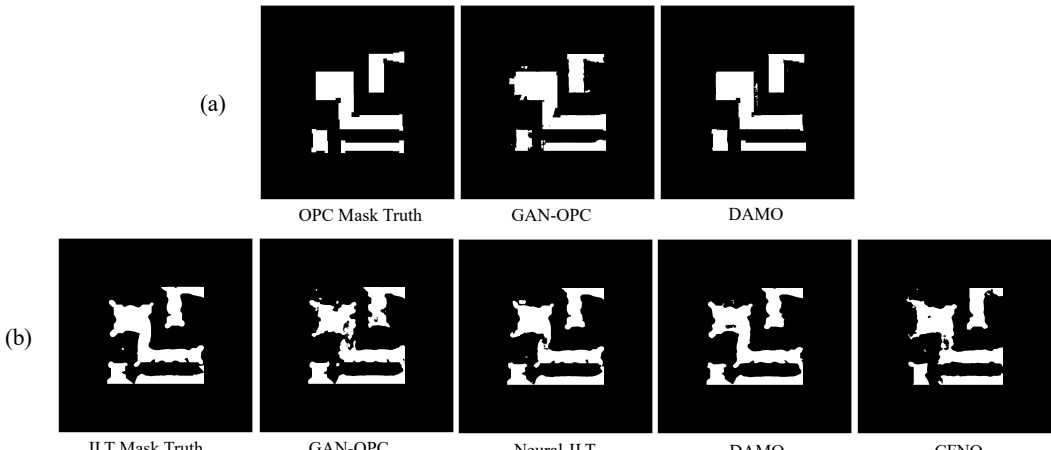

Figure 7: Mask generation examples. (a)Model-based OPC,(b)ILT.

Table 2: Input Ablation Study.

| Metric | Model-based OPC | | ILT | |
|---|---|---|---|---|
| | Metal-layer | Via-layer | Metal-layer | Via-layer |
| $\Delta L_2$ | 1810 ↓ | 134 ↑ | 971 ↑ | 210 ↑ |
| $\Delta$ EPE | 0.04 ↑ | 0.02 ↑ | 1.5 ↓ | 0.1 ↑ |
| $\Delta$ PVB | 833.6 ↓ | 105 ↑ | 104 ↑ | 200 ↑ |
| $\Delta$ Shot | 123.6 ↑ | 19 ↑ | 103 ↑ | 159 ↑ |

Overall, DAMO achieves the lowest $L_2$ and EPE across both layers, demonstrating superior mask generation fidelity, but it sacrifices manufacturability by producing more complex shapes that lead to higher shot counts. Neural-ILT delivers moderate performance across all metrics, balancing accuracy and complexity without clear dominance. By contrast, OPC-GAN achieves the lowest shot counts across both layers, indicating its tendency to generate simpler patterns at the cost of higher $L_2$ and EPE, reflecting its limited ability to capture fine-grained mask details. These results highlight distinct trade-offs among baseline models.

**Input Ablation.** In this section, we examine the importance of the cell tag as an input for mask generation. To assess its impact, we remove the cell tag and retrain the GAN-OPC model reported in Table 1, with the resulting differences in evaluation metrics summarized in Table 2.

The results show that for via-layers, cell information plays a critical role in mask optimization, as removing the cell tag input consistently degrades performance across all metrics. For ILT of metal and via layer, performance also degrades in most metrics when cell tags are absent. Overall, these results highlight the necessity of incorporating cell tags for accurate mask optimization.

## 5 CONCLUSION

In this work, we introduced MaskOpt, a large-scale benchmark dataset for IC mask optimization that captures the hierarchical and context-aware characteristics of real circuit layouts. By curating layout–mask pairs from real designs at the standard-cell level and incorporating variable context windows, MaskOpt offers a faithful representation of practical OPC and ILT scenarios, going beyond the limitations of existing synthetic datasets. Our experiments highlight the critical importance of both cell-aware and context-aware inputs, underscoring their role in improving the fidelity and manufacturability of mask generation. We believe MaskOpt will serve as a valuable resource to advance machine learning methods for efficient and reliable IC mask optimization.

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
