# OpenReview forum: "MaskOpt: A Large-Scale Mask Optimization Dataset to Advance AI in Integrated Circuit Manufacturing"
_ICLR.cc/2026/Conference — ICLR 2026 Conference Withdrawn Submission_

### Official Review · Reviewer_wh2C · 2025-10-25

**Soundness:** 2
**Presentation:** 3
**Contribution:** 2
**Rating:** 4
**Confidence:** 3

**Summary:**

This paper presents a benchmark dataset containing 104714 metal-layer tiles and 121952 via-layer tiles using five designs from the OenROAD flow for IC mask optimization. It focuses on generating the mask at a cell level and provides results on model-based OPC and ILT tasks to highlight the importance of cellaware and context-aware inputs.

**Strengths:**

1. Clear presentation. This paper clearly introduces the background of optical lithography and other concepts.
2. The provided dataset contains much more data compared to the previous benchmark dataset. And it is closer to a real application without using synthetic tiles.
3. This benchmark enables cell-based hierarchical OPC, which provides more information for mask optimization.

**Weaknesses:**

1. This paper doesn’t clearly explain the advantages and weaknesses of modelbased OPC and ILT. Experimental results show some optimization preferences (e.g., methods belonging to model-based OPC tend to optimize the L2 metric, while methods from ILT show greater improvement in EPE), but these are not clearly explained.
2. Fewer baseline methods are included for comparison. The experiment only adopts four baseline methods. For OPC mask prediction, Neural-ILT and CFNO are adaptable but not included for comparison.
3. The mask ground truths are generated using OpenILT platform, and all baseline methods are trained to minimize the loss between model outputs and mask ground truths. The square L2 error between the printed wafer images using mask ground truths and the targeted layout image should be provided as an oracle result.

**Questions:**

1. What are the differences in advantages and weaknesses between modelbased OPC and ILT methods?
2. Can more methods be added to the experimental comparison and analyzed in more detail?
3. Can the mask ground truths generated in this paper produce the closest result to the target layout images? Is minimizing the error with mask ground truths the optimal optimization objective?

---

### Official Review · Reviewer_D1GT · 2025-10-28

**Soundness:** 1
**Presentation:** 3
**Contribution:** 1
**Rating:** 2
**Confidence:** 5

**Summary:**

This paper proposed a lithography benchmark targeting OPC and ILT (pixelated mask optimization). The overall contribution includes larger-scale real design synthesized under NanGATE45 PDK and standard cell-based clips. Its interesting to see efforts contribute to the AI for computational lithography. But I do have concerns about the practicality and impacts of such benchmark compared to prior works. Please see my detailed comments below.

**Strengths:**

-Benchmarks include real chip designs synthesized on open source PDK. Especially, metal layers are more close to true design patterns compared to the previous LithoBench, where metal layer designs are mostly synthetic data.

-Multiple ML solutions are evaluated on the newly introduced benchmark.

-The amount of data is significantly larger than existing benchmarks, potentially benefits ML-based solutions.

**Weaknesses:**

I do have several concerns and comments on the benchmark.
1. Author mentioned the clipping is based on standard cells (plus different sizes of surrounding context), I don't know the motivation to do that: a. STC patterns are usually quite limited, for each technology node, there are typically larger hundreds or 1K STCs. based on this, I really doubt the diversity and coverage of the benchmark. b. If the clips are with lots of similar patterns with different surrounding contexts, this will only pose significantly challenge on ML training. c. The clip has 512x512 nm core region on 1024x1024 tile. However, as far as I know, there are many STCs in NanGate45 that have much larger size than 1024x1024. In this sense, the benchmark does not show advantage over LithoBench.

2. Again for the tile size, its too small for practical usage. In industry practical, the tile size should be at least 8um X 8um or even larger. Models supporting such small clips are typically useless. Even for lithobench, the clip size is already 2umX 2um.

3. In computationaly lithography or precisely ILT, ML methods/genAI methods only serve as add-on for traditional solver because one need to go through at least few iterations of numerical solver for refinement check. However, the experiments does not show any SOTA numerical ILT results like: MultiILT [Sun+, DAC'23], DiffOPC [Chen+, ICCAD24], CurvyILT [Yang+, ISPD'25]

4. By looking at the benchmark details, for example in Fig. 6, some of the patterns are intentionally cut into clips, resulting in patterns cut into half, yielding even much smaller CD than the layer rules. Thus I would say some of the data points do not make any sense.

**Questions:**

Please refer to the weakness.

---

### Official Review · Reviewer_pGry · 2025-10-30

**Soundness:** 2
**Presentation:** 3
**Contribution:** 1
**Rating:** 2
**Confidence:** 4

**Summary:**

The paper introduces MaskOpt, a new dataset for learning-based mask optimization in optical lithography.
The dataset is built from “five real IC designs at the 45nm technology node,” and contains 104,714 metal-layer tiles and 121,952 via-layer tiles. For each tile, the authors provide (1) a clipped layout “core” region aligned with a standard-cell placement, (2) optional extended spatial context around that core (0–128 nm), (3) the associated standard-cell tag, and (4) ground-truth model-based OPC and ILT masks generated using OpenILT. The task is: given layout + context + cell tag, predict either the OPC-style or ILT-style corrected mask.
They benchmark several published deep-learning baselines (GAN-OPC, DAMO, Neural-ILT, CFNO) and evaluate them.
The main claims are: (1) MaskOpt is more realistic than prior datasets such as LithoBench because it preserves standard-cell hierarchy and surrounding context; (2) including standard-cell identity and local context improves mask prediction; and (3) different baseline models expose different trade-offs between fidelity and manufacturability.

**Strengths:**

1. The motivation is clear. Mask optimization is indeed a core bottleneck in modern lithography because it requires repeated forward lithography simulation and careful process-window optimization.

2. The dataset is not just random image crops: tiles are explicitly aligned with standard-cell placements.

**Weaknesses:**

1. The contribution is incremental and the novelty is overstated. The core claimed contribution is “a large-scale dataset for mask optimization that captures cell hierarchy and context.”
But LithoBench already provides >100k labeled via tiles plus tens of thousands of metal-layer tiles, including both target and optimized masks.
Relative to that, MaskOpt differs mainly in (1) how the clips are cropped with different margins, (2) adding a per-sample "cell tag" channel", which is more like a incremental variant of LithoBench rather than a new benchmark.
Besides, all evaluation toolkits are provided by OpenILT and open-sourced model, which further limits the contribution.

2. The benchmark evaluation methodology assumes mask quality is best assessed from a tile's core region after simulating a larger window with an added margin.
This standard trick minimizes boundary artifacts in the core. The paper claims larger margins inherently improve fidelity by providing more context.
However, this is misleading and cannot actually scale to production, as the usable margin is physically limited in manufacturing, and the "center-is-better" bias does not scale to larger or full-chip layouts with cut-off contexts.
Thus, the reported gains are partly an artifact of the windowed simulation, overstating the model's practical performance in full-chip mask synthesis.

2. The paper repeatedly describes the source designs as “five real IC designs at the 45nm node.” In reality, these are designs implemented using the fully open-source OpenROAD flow and the public Nangate45 library, not proprietary cutting-edge designs.
OpenROAD + Nangate45 is great for research, but it is far from manufacturing signoff at 45 nm.

3. The cell tag is injected as a one-hot channel broadcast to 1024×1024 and concatenated with the image.
That means the model can trivially memorize a per-cell lookup table for mask style, which is the opposite of generalization.
The authors do not evaluate generalization to unseen cells, unseen designs, or process drifts.

4. The OpenILT provides several ILT algorithm, and the author didn't tell which ILT solutions they used.

**Questions:**

See Weaknesses

---

### Note · Authors · 2025-11-22

I have read and agree with the venue's withdrawal policy on behalf of myself and my co-authors.